# Low C/N Ratios Promote Dissimilatory Nitrite Reduction to Ammonium in *Pseudomonas* *putida* Y-9 under Aerobic Conditions

**DOI:** 10.3390/microorganisms9071524

**Published:** 2021-07-17

**Authors:** Xuejiao Huang, Wenzhou Tie, Deti Xie, Zhenlun Li

**Affiliations:** 1College of Agronomy, Guangxi University, Nanning 530004, China; twenzhou@163.com; 2Chongqing Key Laboratory of Soil Multiscale Interfacial Process, Southwest University, Chongqing 400716, China; xdt@swu.edu.cn

**Keywords:** *Pseudomonas* *putida*, denitrification, dissimilatory nitrite reduction to ammonium, aerobic conditions, environmental factors

## Abstract

The biogeochemical consequences of denitrification and dissimilatory nitrate reduction to ammonium (DNRA) have a significant influence on nitrogen (N) cycling in the ecosystem. Many researchers have explored these two pathways in soil and sediment ecosystems under anaerobic conditions. However, limited information is available regarding the influence of external environmental conditions on these two pathways in a well-defined experimental system under aerobic conditions. In this study, the impacts of the external environmental factors (carbon source, C/N ratio, pH, and dissolved oxygen) on nitrite reduction through the denitrification and DNRA routes in *Pseudomonas* *putida* Y-9 were studied. Results found that sodium citrate and sodium acetate favored denitrification and DNRA, respectively. Furthermore, neutral pH and aerobic conditions both facilitated DNRA and denitrification. Especially, low C/N ratios motivated the DNRA while high C/N ratios stimulated the denitrification, which was opposite to the observed phenomena under anaerobic conditions.

## 1. Introduction

Nitrate (NO_3_^−^), a mobile anion, is easy to leach from agricultural land and runoff into surface waters, which not only decreases the nitrogen fertilizer efficiency but triggers environmental problems [1,2,3]. Many researchers find that NO_3_^−^ could be removed by microorganisms through assimilatory and dissimilatory processes. Dissimilatory nitrate reduction contains two different pathways: denitrification (NO_3_^−^ → NO_2_^−^ → NO → N_2_O → N_2_) and dissimilatory nitrate reduction to ammonium (DNRA, NO_3_^−^ → NO_2_^−^ → NH_4_^+^) [4,5]. DNRA is a pathway that shares the NO_3_^−^ to nitrite (NO_2_^−^) reaction step with denitrification but reduces NO_2_^−^ to ammonium (NH_4_^+^) [6,7]. The relative proportion of denitrification and DNRA plays a crucial role in nitrogen retention in ecosystems [8].

At present, many studies focus on the environmental factors affecting denitrification versus DNRA given their biogeochemical consequences [9,10,11]. The factors that influence denitrification and DNRA include oxygen concentration [11,12], carbon source [9], and C/N ratio [10,13]. Moreover, the C/N ratio is a significant factor on the above two pathways under the same oxidation status. Usually, denitrification occurs under low C/N ratios, whereas DNRA prevails under high C/N ratios [9,14]. However, these studies were all conducted in mixed systems (such as soil and sediment) under anaerobic conditions.

Our previous study clarified that both denitrification and DNRA exist in *Pseudomonas putida* Y-9 under aerobic conditions by genome-wide and gene knockout as well as ^15^N isotope technologies [15]. Given the scarcity of data available to assess the effect of the environmental factors on denitrification and DNRA in a well-defined experimental system, the object of this study is to explore the external factors (carbon source, C/N ratio, pH, and dissolved oxygen) that determine the denitrification and DNRA fate in strain Y-9.

## 2. Materials and Methods

### 2.1. Microorganism and Culture Media

*P. putida* Y-9 (Genbank No. KP410740), conducting both denitrification and DNRA under aerobic conditions [15], was used in the study.

Luria-Bertani (LB) medium, used as an enrichment medium for the strain contained (per liter, pH = 7.2): 10.0 g tryptone, 5.00 g yeast extract, and 10.0 g NaCl. A denitrification medium (DM) was applied to assess the denitrification and DNRA ability of the strain. It comprised (per liter, pH = 7.2) 7.00 g K_2_HPO_4_, 3.00 g KH_2_PO_4_, 5.13 g CH_3_COONa, 0.10 g MgSO_4_·7H_2_O, 0.49 g NaNO_2_, and 0.05 g FeSO_4_·7H_2_O, yielding an initial NO_2_^−^ concentration around 100 mg/L. The mediums were autoclaved for 30 min at 121 °C.

### 2.2. Evaluation of the Effects of Chloramphenicol on the Growth and Nitrite Reduction Performance of Strain Y-9

Strain Y-9 could perform denitrification and DNRA under aerobic conditions [15]. However, under aerobic conditions, the assimilation of NO_3_^−^ in strain Y-9 is also a major metabolic pathway [16], which might disturb the study of denitrification and DNRA in strain Y-9. However, when using nitrite as substrate, there was nearly no assimilation observed of NO_2_^−^ in the short time (1 h) whereas the accumulation of NH_4_^+^ reached 1.90 mg/L [15]. Therefore, NO_2_^−^ instead of NO_3_^−^ was used as a substrate to explore the fate of denitrification and DNRA by strain Y-9 in this study [8]. The influence of chloramphenicol (200 mg/L) (which could inhibit the bacteria growth) [17] on nitrite conversion of strain Y-9 was also studied here. The purpose was to make sure that only denitrification and DNRA existed during the whole incubation period. A single colony of the isolated strain was cultivated for 36 h in 100 mL LB medium at 15 °C and 150 rpm. After that, 8 mL culture was sampled and centrifuged (4000 rpm, 8 min). The pellet was washed with sterilized water and then inoculated into the DM medium with or without chloramphenicol. The culture conditions were all at 15 °C, 150 rpm in the rotary shaker.

### 2.3. Single-Factor Affecting Dissimilatory Nitrite Reduction

The preserved strain Y-9 was first activated in 100 mL LB medium at 150 rpm and 15 °C for 36 h. Then, the bacterial suspension was inoculated into a 100 mL DM medium containing 0.61 g NaNO_3_ instead of 0.49 g NaNO_2_. Finally, the cells in the logarithmic growth phase were inoculated into a 100 mL DM medium. Chloramphenicol (200 mg/L) was added to the DM medium to inhibit the strain growth. In carbon source experiments, sodium acetate, glucose, sucrose, and sodium citrate were added into a 100 mL DM medium, respectively. The C/N ratio, pH, and shaking speed were 15, 7, and 150 rpm, respectively. To assess the influence of C/N ratios, a 100 mL DM medium was amended with varying concentrations of sodium acetate to make the C/N ratio 2, 4, 8, 15, and 30. The pH and shaking speed were 7 and 150 rpm. For the pH experiments, the initial pH was adjusted by NaOH and HCl to 4, 6, 7, 8, and 9. The carbon source was sodium acetate, the C/N ratio and the shaking speed were 4 and 150 rpm. DO effects were examined by cultivating strain Y-9 under the shaking speeds of 0, 50, 100, and 150 rpm [18,19]. The carbon source was sodium acetate, the C/N ratio and pH were 4 and 7. The cultures were incubated at 15 °C for 4 h. All experiments were carried out in triplicate.

### 2.4. Analytical Methods

During incubation, samples were collected from the cultures to determine the optical density (OD_600_), pH, and nitrogen values. OD_600_ was quantified using a spectrophotometer. The pH value was detected with a pH electrode. The nitrogen values were determined according to the guidelines set by the State Environmental Protection Administration of China [20]. The suspension was used to measure the TN. The supernatant was collected to detect the NH_4_^+^, NO_3_^−^, and NO_2_^−^ concentrations after the sample was centrifuged at 8000 rpm for 5 min. The removal efficiency of TN and NO_2_^−^ was calculated as follows: *R*v = (A − B)/A × 100%, where *R*v is the removal efficiency of TN and NO_2_^−^, while A and B are the initial and final TN and NO_2_^−^ concentrations in the cultures, respectively.

### 2.5. Statistical Analysis and Graphical Work

SPSS Statistics 22 was selected to perform a one-way analysis of variance (ANOVA). Differences at *p* < 0.05 were considered to be statistically significant. Origin 8.6 was employed to perform the graphical work.

## 3. Results

### 3.1. Effect of Chloramphenicol on Strain Growth and Nitrite Reduction

The results of the effect of chloramphenicol on the nitrite reduction by stain Y-9 are shown in Figure 1. The OD_600_ and the concentration of NO_2_^−^ increased and decreased, respectively, in the medium without chloramphenicol (Figure 1A,B). The NH_4_^+^ concentration in the medium without chloramphenicol increased and then decreased, which shows that the NH_4_^+^ was absorbed by strain Y-9 for growth once it was produced (Figure 1C). The concentration of NH_4_^+^ increased during the cultivation period in the chloramphenicol-containing medium (Figure 1C). The reduction of NO_2_^−^ in the chloramphenicol-containing medium was lower than that in no chloramphenicol medium (Figure 1B), and strain Y-9 did not grow (Figure 1A). The above phenomena indicated that chloramphenicol inhibited the growth of strain Y-9, which was identical to the findings of Tiedje et al. [17], thus avoiding the assimilation of the generated NH_4_^+^ by the strain Y-9. Therefore, chloramphenicol (200 mg/L) was added into the DM medium to inhibit the growth of the isolated strain in further study.

### 3.2. The Influence of Carton Source on Dissimilatory Nitrite Reduction

Due to the importance of the carbon source for nitrogen transformation, experiments were performed to quantify the influence of sodium acetate, glucose, sucrose, and sodium citrate on denitrification and DNRA by stain Y-9. Figure 2 illustrates that denitrification and DNRA seem to be affected by different carbon sources. Among all the tested carbon sources, maximum and minimum ammonium production was found when using sodium acetate and glucose as the sole carbon source, respectively (Figure 2C), indicating that sodium acetate was the best carbon source for DNRA by stain Y-9 under aerobic conditions. The OD_600_ in different carbon source mediums was maintained at 0.5–0.6 during the incubation period by adding the chloramphenicol (Figure 2A). The pH decreased when using glucose as a carbon source while being rather stable in the other three carbon sources mediums (*p* < 0.05) (date no shown). These phenomena illustrated that the effect of glucose on strain Y-9 was different from the other three carbon sources. The TN in the sucrose medium hardly changed during the whole incubation process. However, the TN in other carbon sources medium decreased after 4 h of incubation (Figure 2D). He et al. [21] have shown that *P. putida* Y-9 could remove higher TN when using sodium citrate as a carbon source than other carbon sources in a low concentration nitrite medium (15 mg/L NO_2_^−^). Sodium citrate as a carbon source also contributed to the highest TN reduction rate in a high concentration nitrite medium (100 mg/L NO_2_^−^) in our study.

### 3.3. The Influence of C/N Ratio on Dissimilatory Nitrite Reduction

The results of the influence of the C/N ratio on aerobic dissimilatory nitrite reduction by strain Y-9 are shown in Figure 3. During the incubation, the OD_600_ in different C/N ratios medium was maintained at 0.5–0.6 except for that in a medium with a C/N ratio of 2 (Figure 3A). High carbon concentration seems to stimulate the decline of TN, and the reduction efficiency reached to highest (0.77%) at the C/N ratio of 30 (Figure 3D), which illustrated that a high C/N ratio could stimulate the TN removal by strain Y-9 under aerobic conditions. NO_2_^−^ and NH_4_^+^ all decreased and increased under different C/N ratios, respectively (Figure 3B,C). However, the production of NH_4_^+^ reached its maximum at C/N ratios of 4, and reduced when the C/N ratio exceeds 4 (Figure 3C). The result was similar to the report of Zhao et al. [22], who found that the optimal C/N ratio for ammonium production during the nitrate reduction process of *Pseudomonas stutzeri* strain XL-2 using sodium acetate as carbon under aerobic conditions was 5, the production amount of ammonium decreased when C/N ratio exceeded 5. These phenomena implied that the transformation of NO_2_^−^ to NH_4_^+^ might be favored when the supply of carbon was higher.

### 3.4. The Influence of Initial pH on Dissimilatory Nitrite Reduction

The effect of initial pH on dissimilatory nitrite reduction of strain Y-9 was studied and the results are shown in Figure 4. He et al. [21] reported that the denitrification of *P. putida* Y-9 would be inhibited when the pH in a low concentration nitrite medium (15 mg/L) was lower than 6. There was also no TN reduction under acidic conditions (the initial pH was 4 and 6) in high nitrite medium (100 mg/L NO_2_^−^) in our study, which indicated that acidic conditions were not favorable for TN removal by stain Y-9 (Figure 4D). NO_2_^−^ and NH_4_^+^ decreased and increased under different initial pH conditions (except for 4), respectively (Figure 4B,C). The highest production of NH_4_^+^ was found at the initial pH of 7, which suggested that the initial pH of 7 favors the DNRA.

### 3.5. The Influence of Dissolved Oxygen on Dissimilatory Nitrite Reduction

The OD_600_ was maintained at 0.5–0.6 for 4 h by adding the chloramphenicol (Figure 5A). Higher shaking speed stimulated the decline of TN, and the reduction reached the highest at the shaking speed of 100 rpm (Figure 5D). He et al. [21] have found that sufficient DO promote the removal of TN by strain Y-9 in a low concentration nitrite medium (15 mg/L NO_2_^−^), the removal efficiency reached to highest (77.13%) after incubating for 48 h at the shaking speed of 100 rpm while decreased when the shaking speed was higher than 100 rpm. The same phenomenon was observed in our study (Figure 5D), which indicated that the TN removal capacity of strain Y-9 might not be favored above a DO concentration. NO_2_^−^ and NH_4_^+^ all decreased and increased under different shaking speeds, respectively (Figure 5B,C). The higher production rate of NH_4_^+^ was detected at a shaking speed of 150 rpm.

## 4. Discussion

The nitrate can be transformed to gaseous nitrogen through the denitrification (NO_3_^−^ → NO_2_^−^ → NO → N_2_O → N_2_) and to ammonium through DNRA (NO_3_^−^ → NO_2_^−^ → NO → NH_4_^+^, or NO_3_^−^ → NO_2_^−^ → NH_4_^+^) [22,23,24]. During the process of DNRA, NirK/NirS could catalyze NO_2_^−^ into NO, while NirB/NirC/CysG could catalyze NO_2_^−^ directly into NH_4_^+^. In our previous study, strain Y-9 was found to reduce nitrate to gas nitrogen, and conduct the DNRA process under aerobic conditions according to the ^15^N isotope test. The gene *nirBD* relative to DNRA was found during nitrogen metabolism by Kyoto Encyclopedia of Genes and Genomes (KEGG), which indicated that the DNRA pathway in strain Y-9 was NO_3_^−^ → NO_2_^−^ → NH_4_^+^, and the decrease of nitrogen in the system was owing to the denitrification conducted by the isolate [15]. In summary, strain Y-9 can conduct both denitrification and DNRA under aerobic conditions.

The influence of carbon sources, C/N ratio, pH, and DO on the denitrification and DNRA conducted by strain Y-9 were assessed in this study. The results found that these two pathways were markedly affected by different carbon sources. Sodium citrate and sodium acetate was the best carbon source for denitrification and DNRA of strain Y-9 under aerobic conditions, respectively. The acidic environment was not beneficial to the denitrification and DNRA of strain Y-9. The adaptation to pH of DNRA was better than that of denitrification, which was consistent with the previous study [25,26,27]. Traditionally, denitrification was believed to be a strictly anaerobic process [28]. But now, many researchers had found that denitrification could also be conducted under aerobic conditions [29,30,31]. Strain Y-9 used in this study proved to conduct the denitrification under aerobic conditions [15,32], which was consistent with our research results. Further, most of the scholars reported that DNRA happens under anaerobic conditions [5,33,34,35]. This study found that oxygen had a stimulative effect on DNRA and could promote the reduction of NO_2_^−^ to NH_4_^+^ for *P. Putida* Y-9 strain.

Many studies have reported that denitrification dominated at low C/N ratios, whereas ammonium was the predominant product at high C/N ratios under anaerobic conditions [9,36,37,38]. The reason is that 1 mol NO_2_^−^ receives more electrons (6 mol) during the process of DNRA than that during denitrification (3 mol), thus DNRA tends to occur in high C/N ratios environments [39,40]. Interestingly, our study found that low C/N ratios promote DNRA while high C/N ratios stimulate denitrification in aerobic conditions, which was contrary to the phenomenon under anaerobic conditions, the reason of the difference must be further tested.

## Figures and Tables

**Figure 1 microorganisms-09-01524-f001:**
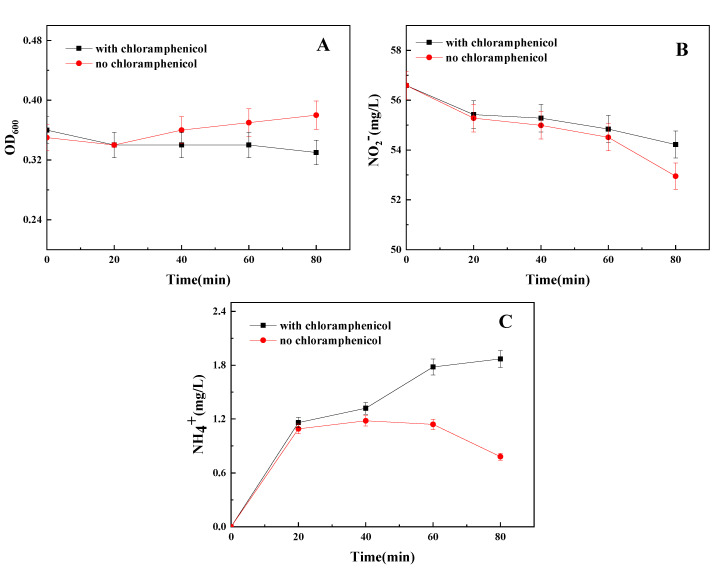
Influence of chloramphenicol on OD_600_ (**A**), NO_2_^−^ (**B**), and NH_4_^+^ (**C**) by strain Y-9. Values are means ± SD for triplicates.

**Figure 2 microorganisms-09-01524-f002:**
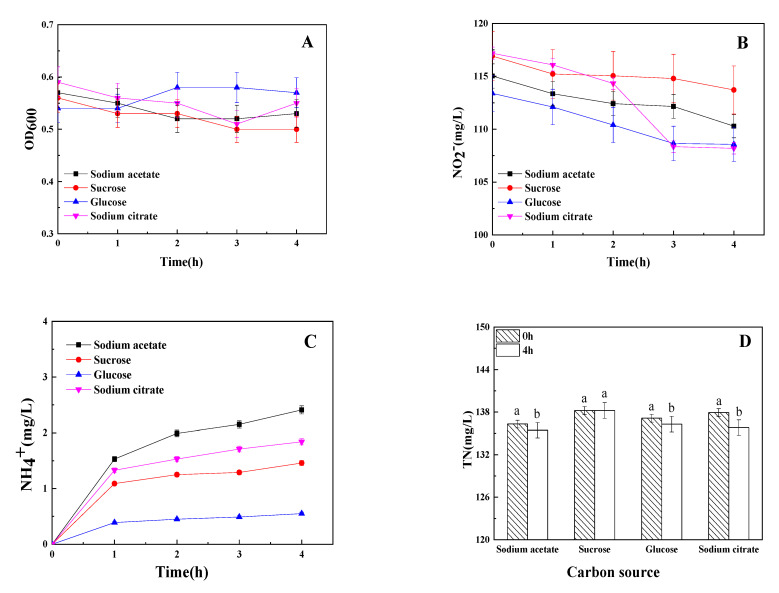
Influence of carbon source on OD_600_ (**A**), NO_2_^−^ (**B**), NH_4_^+^(**C**), and TN (**D**) by strain Y-9. Values are means ± SD for triplicates. Different letters indicate significant differences between treatments at *p* < 0.05.

**Figure 3 microorganisms-09-01524-f003:**
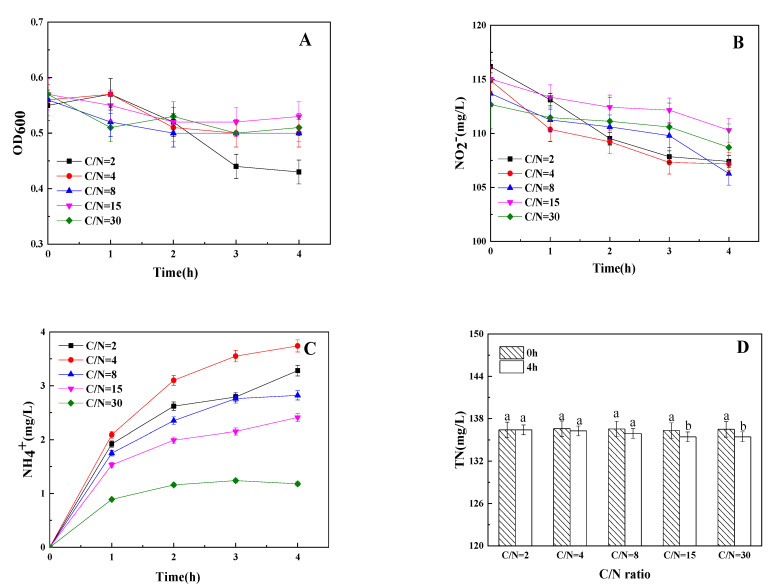
Influence of C/N ratio on OD_600_ (**A**), NO_2_^−^ (**B**), NH_4_^+^ (**C**), and TN (**D**) by strain Y-9. Values are means ± SD for triplicates. Different letters indicate significant differences between treatments at *p* < 0.05.

**Figure 4 microorganisms-09-01524-f004:**
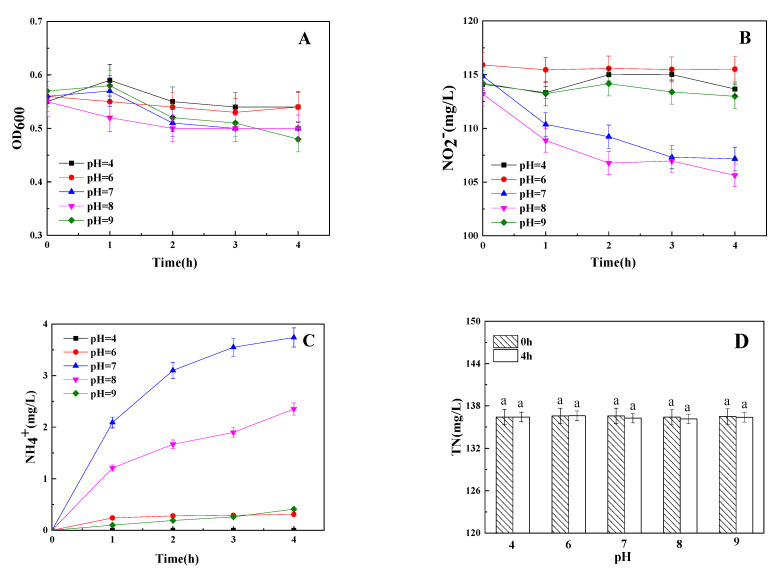
Influence of pH on OD_600_ (**A**), NO_2_^−^ (**B**), NH_4_^+^ (**C**), and TN (**D**) by strain Y-9. Values are means ± SD for triplicates. Different letters indicate significant differences between treatments at *p* < 0.05.

**Figure 5 microorganisms-09-01524-f005:**
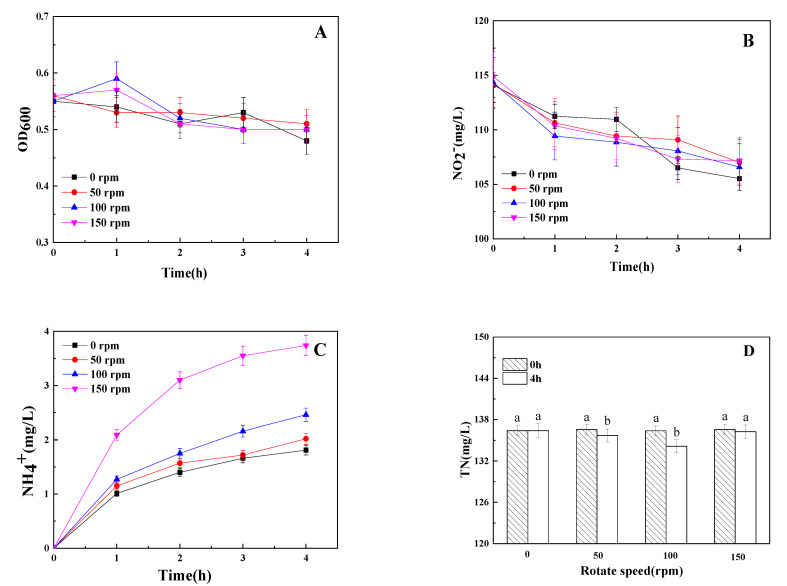
Influence of DO on OD_600_ (**A**), NO_2_^−^ (**B**), NH_4_^+^ (**C**), and TN (**D**) by strainY-9. Values are means ± SD for triplicates. Different letters indicate significant differences between treatments at *p* < 0.05.

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
