# Peer review of "Low C/N Ratios Promote Dissimilatory Nitrite Reduction to Ammonium in Pseudomonas putida Y-9 under Aerobic Conditions"

_microorganisms, 2021, doi:10.3390/microorganisms9071524_

Round 1
Reviewer 1 Report
The manuscript entitled "Low C/N ratios promote dissimilatory nitrite reduction to ammonium in Pseudomonas putida Y-9 under aerobic conditions" presents the results from flasks experiments of Pseudomonas putida Y-9 strain to study two reduction processes of nitrite, namely denitrification and DNRA. The manuscript content is within the topic of Microorganisms journal and the presented work is original to the reviewer’s best knowledge. The experimental procedure is logical and the presented results are interesting. The quality of the English language is moderate and should be improved. The presentation of the results and the experimental procedure can be improved to facilitate the reader. Some arguments and conclusions are not adequately justified and should be rephrased to be accurate.
Concerning the presentation of the manuscript the authors did not follow the template of the journal; especially, the lack of line numbers made the reviewing process more difficult.
More specifically:
- Abstract: “…consequences for N retention in the system.” and “…two pathways in the mixed system under anaerobic conditions.”, which system do the authors refer to?
- Abstract: “In this study, the impacts……were explored”. A more clear formulation can be used, i.e. “In this study, the impacts of external environmental conditions on nitrite reduction through the denitrification or DNRA route in Pseudomonas putida Y-9 were studied”.
- Page 3: The description of the experimental procedure should be more detailed and clearer. The authors provide experimental conditions in the results sections, for example, at the start of chapters 3.3 and 3.4, these parts should be included in chapter 2. Chapter 2 contains a lot inconsistent descriptions. For example the authors mention the LB medium (enrichment medium) without referring to how they used it. The DM medium refers that the concentration of nitrite is 0.49g NaNO2, whereas in various parts of the manuscript the authors refer that the NO2 concentration is “110 mg/L” or in another part around 100 mg/L NO2. They use the term basal medium without giving its composition or clarifying to which medium they refer to. The authors are kindly asked to rewrite chapter 2, providing a detailed and clear description of the experimental procedure that would easily allow its replication from other researchers!
- Can the authors provide more information/justification on why they choose to use NO2 instead of NO3 for their test? How the NO2 doesn’t interfere to nitrate assimilation pathway, given that it’s an intermediate product of NO3 assimilation?
- Why did the authors cultivate the strain at “… 150 rpm, C/N ratio of 4, initial pH, and 15 oC for 4 h.”? What’s the purpose of this/these experiment/s?
- The authors refer to the “…decline rate of nitrogen…”. Do they refer to the Total Nitrogen (TN)? Also the parameter that they calculate is not a rate (since it does not describe the temporal evolution of a parameter) but a reduction/removal efficiency. The authors are kindly asked to use the correct term throughout the whole manuscript.
- The manuscript lacks of a proper statistical analysis of the results. IN many case the differences in the concentration of various nitrogen fractions or of the pH are so small that they may not be statistically important (e.g. Fig. 3D, 3E, 4D, 5E, 5B, etc.). In these cases, the authors should first conclude whether the difference are not due to samples’ variation or analysis errors (e.g. ANNOVA), before claiming that the x-parameter results in increased y-value.
- The reference Van et al., 2017 is not present in the reference list. Moreover, the reference list should be reformulated according to MDPI template.
- Page 5: “However, the TN in other carbon sources medium continuously decreased in 4h”. This statement is not supported by the presented data. TN concentration in the presence of sucrose seems unchanged. The term “continuously” is not also correct given that only one time point (@ 4h) is available/presented.
- Page 6: “Under neutral to alkaline…of initial pH (Fig.4D).” The TN reduction seems insignificant, and it’s much lower than the SD of the measurements. The reviewer assumes that an ANNOVA would show that any differences are not statistically significant, therefore the statement of the authors is not supported from the experimental data.
- Page 8: “The initial habitat…to conduct the process of DNRA”. The authors’ statement, although it seems plausible, is not supported by the presented results. To support this statement the authors should have performed analysis of the expressed genes and/or the protein/enzyme profile under the different conditions that would show that different genes are expressed in the different growth environments.
- Page 9: “It is notable that, low…under anaerobic conditions”. This statement is already stated, there is no reason for its repetition.
- Discussion and conclusions should be merged into one chapter.
- Given the large number of Figures, pH figures does not seem to provide any useful information in Fig. 2, 3 and 5 given that their value remain relatively stable.
The authors are also kindly asked to go through the attached pdf file and edit all the minor grammatically and syntactically issues that are noted by the reviewer.

Author Response
The manuscript entitled "Low C/N ratios promote dissimilatory nitrite reduction to ammonium in Pseudomonas putida Y-9 under aerobic conditions" presents the results from flasks experiments of Pseudomonas putida Y-9 strain to study two reduction processes of nitrite, namely denitrification and DNRA. The manuscript content is within the topic of Microorganisms journal and the presented work is original to the reviewer’s best knowledge. The experimental procedure is logical and the presented results are interesting. The quality of the English language is moderate and should be improved. The presentation of the results and the experimental procedure can be improved to facilitate the reader. Some arguments and conclusions are not adequately justified and should be rephrased to be accurate.
Answer: Thank you for the affirmation of this manuscript. We have modified our manuscript carefully according to your reviews.
Concerning the presentation of the manuscript the authors did not follow the template of the journal; especially, the lack of line numbers made the reviewing process more difficult.
Answer: We are so sorry to confuse you owing to not following the template of the journal. Now, we have modified the article to follow the template of the journal.
More specifically:
- Abstract: “…consequences for N retention in the system.” and “…two pathways in the mixed system under anaerobic conditions.”, which system do the authors refer to?
Answer: “system” refers to “the ecosystem”, we have modified it in the abstract and throughout the text.
- Abstract: “In this study, the impacts……were explored”. A more clear formulation can be used, i.e. “In this study, the impacts of external environmental conditions on nitrite reduction through the denitrification or DNRA route in Pseudomonas putidaY-9 were studied”.
Answer: Thanks for your advice. “In this study, the impacts of the external environments on denitrification and DNRA in Pseudomonas putida Y-9 were explored.” has been changed to “In this study, the impacts of the external environmental factors (carbon source, C/N ratio, pH, and dissolved oxygen) on nitrite reduction through the denitrification and DNRA routes in Pseudomonas putida Y-9 were studied. ”
- Page 3: The description of the experimental procedureshould be more detailed and clearer. The authors provide experimental conditions in the results sections, for example, at the start of chapters 3.3 and 3.4, these parts should be included in chapter 2. Chapter 2 contains a lot inconsistent descriptions. For example the authors mention the LB medium (enrichment medium) without referring to how they used it. The DM medium refers that the concentration of nitrite is 0.49g NaNO2, whereas in various parts of the manuscript the authors refer that the NO2 concentration is “110 mg/L” or in another part around 100 mg/L NO2. They use the term basal medium without giving its composition or clarifying to which medium they refer to. The authors are kindly asked to rewrite chapter 2, providing a detailed and clear description of the experimental procedure that would easily allow its replication from other researchers!
Answer: Thank you for your comment. We have rewritten the experimental procedure. For example, at the start of chapters 3.3 and 3.4, these parts have been included in chapter 2; the application of the LB medium has been added; the component of DM medium has been clarified; etc. The relative modifications have been highlighted by red color.
- Can the authors provide more information/justification on why they choose to use NO2 instead of NO3 for their test? How the NO2 doesn’t interfere to nitrate assimilation pathway, given that it’s an intermediate product of NO3 assimilation?
Answer: The reason why we choose to use NO2- instead of NO3- for the test is “ Strain Y-9 could perform denitrification and DNRA under aerobic conditions [15]. However, the assimilation of NO3- also exists in strain Y-9 under aerobic conditions and occupies the major status[16], which might disturb the analysis of denitrification and DNRA in strain Y-9. However, there was nearly no assimilation observed of NO2- in the short time (1 h) when using nitrite as the substrate, and the accumulation of NH4+ reached 1.90 mg/L[15]. Therefore, NO2- instead of NO3- was used as a substrate to explore the fate of denitrification and DNRA by strain Y-9 in this study [8].”. We have added the sentence to the article.
- Why did the authors cultivate the strain at “… 150 rpm, C/N ratio of 4, initial pH, and 15 oC for 4 h.”? What’s the purpose of this/these experiment/s?
Answer: Thank you for your question. The sentence is not accurate and we have modified it in the article. We have provided a detailed and clear description of the experimental procedure in the article. We hope you understand.
- The authors refer to the “…decline rate of nitrogen…”. Do they refer to the Total Nitrogen (TN)? Also the parameter that they calculate is not a rate (since it does not describe the temporal evolution of a parameter) but a reduction/removal efficiency. The authors are kindly asked to use the correct term throughout the whole manuscript.
Answer: Thank you for your comment. “…decline rate of nitrogen…” refers to “TN and NO2- ”, we have modified it in the article. Moreover, the wrong word “rate” has been changed to “efficiency” throughout the whole manuscript.
- The manuscript lacks of a proper statistical analysis of the results. IN many case the differences in the concentration of various nitrogen fractions or of the pH are so small that they may not be statistically important (e.g. Fig. 3D, 3E, 4D, 5E, 5B, etc.). In these cases, the authors should first conclude whether the difference are not due to samples’ variation or analysis errors (e.g. ANNOVA), before claiming that the x-parameter results in increased y-value.
Answer: Thanks for your suggestion. The data has been analyzed using ANOVA. We have modified the relative descriptions in the article and Figures.
- The reference Van et al., 2017 is not present in the reference list. Moreover, the reference list should be reformulated according to MDPI template.
Answer: Thanks for your advice. We have added the reference Van et al., 2017 in the reference list. Further, the reference list has been reformulated according to MDPI template.
- Page 5: “However, the TN in other carbon sources medium continuously decreased in 4h”. This statement is not supported by the presented data. TN concentration in the presence of sucrose seems unchanged. The term “continuously” is not also correct given that only one time point (@ 4h) is available/presented.
Answer: Thanks for your comment. The sentence “The TN in glucose medium hardly changed during the whole incubation process. However, the TN in other carbon sources medium continuously decreased in 4 h” has been changed to “ The TN in sucrose medium hardly changed during the whole incubation process. However, the TN in other carbon sources medium decreased after 4 h of incubation”.
- Page 6: “Under neutral to alkaline…of initial pH (Fig.4D).” The TN reduction seems insignificant, and it’s much lower than the SD of the measurements. The reviewer assumes that an ANNOVA would show that any differences are not statistically significant, therefore the statement of the authors is not supported from the experimental data.
Answer: Thank you for your comment. We have done the ANOVA and modified Fig.4D. Further, the sentence “Under neutral to alkaline…of initial pH (Fig.4D).” has been deleted since it is not suitable in the chapter.
- Page 8: “The initial habitat…to conduct the process of DNRA”. The authors’ statement, although it seems plausible, is not supported by the presented results. To support this statement the authors should have performed analysis of the expressed genes and/or the protein/enzyme profile under the different conditions that would show that different genes are expressed in the different growth environments.
Answer: Thank you very much for your kind advice. The relative sentences “The initial habitat of strain Y-9 used in this experiment was in low dissolved oxygen content all year round, but the separation process and purification process of the strain were completed in the aerobic environment. It was speculated that these reasons caused the low sensitivity of strain Y-9 to dissolved oxygen. Anaerobic and micro-aerobic conditions which were similar to the original habitat were no longer favorable for the strain Y-9 to conduct the process of DNRA.” has been deleted.
- Page 9: “It is notable that, low…under anaerobic conditions”. This statement is already stated, there is no reason for its repetition.
Answer: Thank you for your comment. The sentence “It is notable that, low C/N ratios led to the predominance of the dissimilatory nitrite reduction to ammonium, while high C/N ratios conducted to the predominance of denitrification, which was contrary to the phenomenon under anaerobic conditions. ” has been deleted.
- Discussion and conclusions should be merged into one chapter.
Answer: The discussion and conclusions have been merged into one chapter.
- Given the large number of Figures, pH figures does not seem to provide any useful information in Fig. 2, 3 and 5 given that their value remain relatively stable.
Answer: Thanks for your suggestion. The Figures about pH have been deleted and the relative modifications have been done throughout the article.
The authors are also kindly asked to go through the attached pdf file and edit all the minor grammatically and syntactically issues that are noted by the reviewer.
Answer: Thank you for your remind. We have gone through the attached pdf file and edited all the minor grammatical and syntactic issues that are noted by the reviewer.
The language of this manuscript has been revised by a native English speaker. With his help, we have checked the manuscript carefully to revise the punctuation marks, grammar, and sentences badly structured in the manuscript. The changes have been highlighted by red color in the article.
Thanks for your reconsideration of our manuscript.
Yours sincerely
Xuejiao Huang
Reviewer 2 Report
The manuscript entitled “Low C/N ratios promote dossimilatory nitrite reduction to ammonium in Pseudomonas putida Y-9 under aerobic conditions” have demonstrated effects of the carbon source, C/N ratio, pH and dissolved oxygen on dissimilatory nitrite reduction.
Authors should correct manuscript according to the suggestion
Minor issues:
Abstract
Abstract should be rewritten. There is no clear purpose, the methods that have been used and there is no concrete research result
Introduction
At the end of introduction Authors should describe in more detail aim of this study
Materials and methods
The main aspect for improvement is statistics. There is no statistical analysis that would make it possible to compare the samples with each other and with the control statistical methods should be added e.g. ANOVA
References
Authors should checked and corrected References according to journal guidelines
Author Response
The manuscript entitled “Low C/N ratios promote dossimilatory nitrite reduction to ammonium in Pseudomonas putida Y-9 under aerobic conditions” have demonstrated effects of the carbon source, C/N ratio, pH and dissolved oxygen on dissimilatory nitrite reduction.
Authors should correct manuscript according to the suggestion
Minor issues:
Abstract
Abstract should be rewritten. There is no clear purpose, the methods that have been used and there is no concrete research result
Answer: Thank you for your suggestion. Abstract has been rewritten carefully.
Introduction
At the end of introduction Authors should describe in more detail aim of this study
Answer: Thank you for your comment. The detailed aim of this study is “ Given the scarcity of data available to assess the effect of the environmental factors on denitrification and DNRA in a well-defined experimental system. Therefore, the object of this study was to explore the external factors (carbon source, C/N ratio, pH, and dissolved oxygen) that determine the denitrification and DNRA fate in strain Y-9 .”. We have added it to the article.
Materials and methods
The main aspect for improvement is statistics. There is no statistical analysis that would make it possible to compare the samples with each other and with the control statistical methods should be added e.g. ANOVA
Answer: Thank you for your suggestion. We have done the ANOVA and added the relative description “SPSS Statistics 22 was selected to perform One-way analysis of variance (ANOVA). Differences at P<0.05 were considered to be statistically significant.” into the manuscript.
References
Authors should checked and corrected References according to journal guidelines
Answer: Thank you for your advice. The References have been checked and corrected according to journal guidelines.
The language of this manuscript has been revised by a native English speaker. With his help, we have checked the manuscript carefully to revise the punctuation marks, grammar, and sentences badly structured in the manuscript. The changes have been highlighted by red color in the article.
Thanks for your reconsideration of our manuscript.
Yours sincerely
Xuejiao Huang
Round 2
Reviewer 1 Report
The authors successfully addressed the reviewer’s comments and the manuscript quality has been significantly improved. Some minor grammatically and syntactically issues are noted by the reviewer in the attached pdf file that should be addressed. The reviewer has also two last comments:
- The last sentence of the manuscript (Line 158) has already been mentioned in Lines 153-155, so there is no need to be repeated.
- The reviewer kindly asks the authors to re-check the ANOVA results for Fig. 3D and 4D. It seems strange that the analysis denotes the differences in TN concentration for C/N = 4 and 8 (Fig. 3D), and for pH = 7 and 8 (Fig. 4D) as significantly different, given their small differences and the comparable SDs...

Author Response
1.The last sentence of the manuscript (Line 158) has already been mentioned in Lines 153-155, so there is no need to be repeated.
Answer: Thank you for your comment. The last sentence of the manuscript (Line 158) has been deleted. Moreover, the last paragraph has also already been mentioned in Lines 140-149. Therefore, we deleted the last paragraph after thinking over it. We hope you understand. If you think it is necessary to use this paragraph, we will add it again.
2.The reviewer kindly asks the authors to re-check the ANOVA results for Fig. 3D and 4D. It seems strange that the analysis denotes the differences in TN concentration for C/N = 4 and 8 (Fig. 3D), and for pH = 7 and 8 (Fig. 4D) as significantly different, given their small differences and the comparable SDs... Answer: Thank you for your kindly remind. We have re-checked the ANOVA results for Fig.3D,4D, and 5D and modified the inappropriate place.
Yours sincerely,
Xuejiao Huang